# Enhancement of Neutralization Responses through Sequential Immunization of Stable Env Trimers Based on Consensus Sequences from Select Time Points by Mimicking Natural Infection

**DOI:** 10.3390/ijms241612642

**Published:** 2023-08-10

**Authors:** Mingming Wan, Xiao Yang, Jie Sun, Elena E. Giorgi, Xue Ding, Yan Zhou, Yong Zhang, Weiheng Su, Chunlai Jiang, Yaming Shan, Feng Gao

**Affiliations:** 1National Engineering Laboratory for AIDS Vaccine, School of Life Sciences, Jilin University, Changchun 130012, China; mingmingw77@163.com (M.W.); yangxiao19900806@163.com (X.Y.); sujie20@mails.jlu.edu.cn (J.S.); dingxue20@mails.jlu.edu.cn (X.D.); julysec@jlu.edu.cn (Y.Z.); zhypharm@jlu.edu.cn (Y.Z.); suweiheng@jlu.edu.cn (W.S.); jiangcl@jlu.edu.cn (C.J.); 2Vaccine and Infectious Disease Division, Fred Hutchinson Cancer Center, Seattle, WA 98109, USA; egiorgi@fredhutch.org; 3Key Laboratory for Molecular Enzymology and Engineering, the Ministry of Education, School of Life Sciences, Jilin University, Changchun 130012, China; 4Institute of Molecular and Medical Virology, School of Medicine, Jinan University, Guangzhou 510632, China; 5Key Laboratory of Viral Pathogenesis & Infection Prevention and Control (Jinan University), Ministry of Education, Guangzhou 510632, China

**Keywords:** HIV-1, vaccine, sequential immunization, consensus, Env trimers

## Abstract

HIV-1 vaccines have been challenging to develop, partly due to the high level of genetic variation in its genome. Thus, a vaccine that can induce cross-reactive neutralization activities will be needed. Studies on the co-evolution of antibodies and viruses indicate that mimicking the natural infection is likely to induce broadly neutralizing antibodies (bnAbs). We generated the consensus Env sequence for each time point in subject CH505, who developed broad neutralization activities, and selected five critical time points before broad neutralization was detected. These consensus sequences were designed to express stable Env trimers. Priming with the transmitted/founder Env timer and sequential boosting with these consensus Env trimers from different time points induced broader and more potent neutralizing activities than the BG505 Env trimer in guinea pigs. Analysis of the neutralization profiles showed that sequential immunization of Env trimers favored nAbs with gp120/gp41 interface specificity while the BG505 Env trimer favored nAbs with V2 specificity. The unique features such as consensus sequences, stable Env trimers and the sequential immunization to mimic natural infection likely has allowed the induction of improved neutralization responses.

## 1. Introduction

Human immunodeficiency virus (HIV) has caused 40.1 million deaths due to the acquired immune deficiency syndrome (AIDS)-related illnesses by 2022 [1]. It has been a grievous global health threat since the start of the pandemic. Although the combination antiretroviral therapy (cART) can effectively suppress HIV replication and has significantly improved the life span and quality of AIDS patients [2,3], long-term medications have side effects and are prone to developing drug resistance. In addition, 1.5 million people are still infected with HIV in 2022 alone [1]. Thus, a preventive vaccine is urgently needed to curb this devastating pandemic [4]. However, all HIV vaccine efficacy trials have failed, except the RV144 trial, which showed a modest efficacy [5,6], partly due to the enormous diversity of the HIV-1 genomes [7]. Therefore, it is most likely that an effective HIV-1 vaccine needs to induce broadly neutralizing Abs (bnAbs) that are able to neutralize diverse HIV-1 variants [8,9].

The envelope glycoprotein (Env) of HIV-1 forms trimers on the surface of the viral membrane and is the sole target for anti-HIV neutralizing antibodies (nAbs) [10,11]. Initial attempts to develop a protective HIV-1 vaccine using monomeric gp120 failed to prevent HIV-1 infection or delay disease progression in the clinical trial [12,13]. Later, more stable and native-like Env trimers are found to be superior for the induction of bnAbs than Env monomers [14,15,16,17]. The BG505.SOSIP trimer has been widely used for various vaccine designs to identify the best approaches for the induction of bnAb responses [18,19]. Later, native flexibly linked (NFL) trimers were found to be able to further increase stability and be applied to more Envs from different HIV-1 strains [20]. More recently, the uncleaved prefusion-optimized (UFO) BG505 trimers showed further improvement in the stability of Env trimers [21], and this was later further optimized by the second generation of UFO Env trimers [22]. More recently, the nanoparticle form of the BGF505 UFO Env trimers showed much improved retention in lymph node follicles, presentation on follicular dendritic cell dendrites and germinal center reactions [23].

Studies on antibody–virus co-evolution in HIV-1-infected individuals who developed broad neutralization activities have demonstrated that bnAbs matured only after extensive diversification of the *env* gene [24,25,26]. Similar results were also observed in long-term-SHIV-infected rhesus macaques [27,28]. This indicates that the continuous diversification of the *env* genes plays an important role in the generation of bnAbs in the infected individuals [24,25]. The emergence of these nAbs drives the viruses to escape from autologous neutralization and to become more diversified, while at the same time, the vast genetic variants drive the continuous maturation of the bnAbs. This antibody–virus arms race can lead to the induction of bnAbs in the same infected individuals [24,29,30,31]. In our previous studies, two bnAb lineages, CH103 and CH253, were isolated from the same HIV-1-infected subject CH505 [24,25]. Interestingly, the neutralization escape mutants selected by one bnAb lineage (CH235) gained higher affinity and neutralization susceptibility to another bnAb lineage (CH103), and thus drove the further maturation of the CH103 bnAb lineage [25]. Taken together, these results show that mimicking the natural virus evolution via sequential immunization with select Env immunogens from different time points during infection is likely to induce broad neutralization activities.

Sequential immunization has been used to develop vaccines against highly variable viruses, such as influenza viruses [32,33] and human papillomaviruses [34]. This approach has also been used for HIV-1 vaccine development, but no significant improvements have been achieved by simply using the unrelated Env immunogens from different virus strains [35,36,37]. Based on the Ab-virus co-evolution theory, the lineage design has been developed for HIV-1 vaccines to elicit broad neutralization activities using sequential immunization of immunogens from the same individuals [38,39]. However, sequential immunization with some natural gp120 Env monomers from HIV-1-infected individual CH505 did not show significantly improved nAb responses [40,41]. In the germline-targeting approach, the precursor B cells can be triggered by computation-guided germline-targeting gp120 outer domain immunogens (eOD-GT) and then sequentially boosted using eOD-GTs with escalating bnAb-binding affinities [42,43]. While eOD-GT8 could successfully trigger germlines and expand the repertoire, broad neutralization against heterologous tier 2 viruses have not been detected [44,45].

Our previous studies showed that the consensus Env sequence of all subtypes among group M viruses not only reduced the genetic distances with all subtypes [46,47,48], but also elicited broad T cell responses and nAb responses in small animals and non-human primates [49,50,51]. The previous studies show that the random selection of natural Env gp120 sequences from different time points may not fully represent the Env quasispecies populations that are responsible for driving bnAb maturation and is likely to skew the immune responses toward particular variants. In this study, we will take advantage of the detailed knowledge of the co-evolution of bnAbs and viruses in subject CH505 and determine whether the highly stable Env trimers generated with consensus Env sequences from different time points can better recapitulate the ability of the Env variants to drive bnAb maturation.

## 2. Results

### 2.1. Generation of the Env Consensus Sequences for Various Time Points from Subject CH505

One advantage of the consensus sequences in vaccine design is that it can reduce the overall genetic differences among analyzed sequences [46,51]. Our previous study showed that the genetic diversification of the *env* gene proceeded the development of neutralization breadth in subject CH505 [25], indicating that increasing viral genetic diversification plays a key role in driving the maturation of bnAbs. Thus, mimicking the natural infection history in CH505 is likely to induce bnAb responses. HIV-1 Env sequences from each time point, from week 4 post-infection up to week 78, when broad neutralization against hard-to-neutralize tier 2 viruses had been detected in subject CH505 [24], were aligned to generate the consensus sequences (Figure A1). Env consensus sequences from weeks 4, 8, 9, 10 and 14 were identical to the transmitted/founder (T/F) Env sequence, whereas the week 7 consensus sequence differed by one amino acid at position 266 (N266K). Env consensus sequences from weeks 20 and 22 differed by three amino acids (aa) and a 3-aa insertion (Figure A1). Env consensus sequences from the week 30, 53 and 78 viruses become increasingly more divergent, especially with insertions in the V1 and V5 regions. All but one aa substitution in gp41 (at weeks 53 and 78) was found in the gp120 region. Therefore, only consensus sequences for weeks 22 (Con_w22), 30 (Con_w30), 53 (Con_w53) and 78 (Con_w78) were selected for the design of Env trimers. A T/F variant (w4.26), which differed from the T/F virus by one amino acid at position 501 (A501V) and showed higher binding to CH103 lineage Abs and higher sensitivity to autologous neutralization [25], was also used to make the Env trimer to test whether it would enhance stimulation of B cell precursors producing CH103- and CH235-like bnAbs.

### 2.2. Characterization of Native-like UFO CH505 Env Trimers

Since stable native-like Env trimers have shown promise to induce nAb responses against tier 2 viruses [52], we designed the consensus *env* genes to express them as stable native-like UFO Env trimers, as previously described [21]. The env gene was modified by replacing its natural signal peptide with a tPA signal peptide, deleting the gp120/gp41 cleavage site by inserting a G4S linker, engineering a disulfide bond (A501C/T605C), generating an I559P mutation to destabilize the post-fusion state and truncating gp41 before the transmembrane domain to express it as the secreted UFO trimers. The final plasmids expressing UFO Env trimers were constructed for T/F, w4.26, Con_w22, Con_w30, Con_w53 and Con_w78 (Figure 1).

Each UFO Env trimer expressing plasmid was transfected into HEK293-6E cells. Three days later, the cell culture supernatants were harvested and Env proteins were purified via affinity chromatography using lectin columns. The purified Env proteins were then separated using size exclusion chromatography (SEC) using Superdex 200 to obtain UFO trimers. The Env trimers were predominant in the most purified preparations using lectin columns, except w4.26 and Con_w53, for which more dimers and monomers were observed (Figure 2a). Blue native gel analysis of collected fractions of each Env protein confirmed that the predominant form was the trimer for T/F, Con_w22, Con_w30, Con_w53 and Con_w78, although more monomers were detected for Con_w53 (Figure 2b and Figure A2). Much fewer trimers were detected for w4.26, in which more dimers, monomers and multimers were present. This was possibly due to the mutation A501V at the UFO disulfide bond site disrupting the conformation on the stable trimer (Figure 1). All UFO Env trimers were confirmed by Western blot analysis (Figure 2c and Figure A3). 

### 2.3. Induction of HIV-1-specific Binding Abs

One genetic variant, w4.26, had a single A501V mutation and bound to the entire CH103 lineage Abs at a higher affinity. It is likely that it could trigger the CH103-like B cell precursors better. Thus, we designed two CH505 Env sequential immunization groups: one primed with the T/F Env and the other one primed with w4.26 Env. Both groups were subsequently immunized with four sequential later time point CH505 consensus UFO Env trimers, Con_w22, Con_w30, Con_w53 and Con_w78, in the same order as they were found in subject CH505 (Figure 3a and Table 1). The positive control group was repeatedly immunized with BG505 UFO trimer five times, and the negative control group was immunized with PBS. There were six guinea pigs in each immunization group.

HIV-1-specific binding antibody titers were assessed with sera collected two weeks post each immunization. After the second immunization, all three immunization groups had good antibody titers, and the antibody responses were boosted after each additional immunization (Figure 3b). Interestingly, the antibody titers were continuously boosted in the w4.26 group after each boost, while they maintained similar levels in the CH505 T/F and BG505 groups after the fourth and fifth boosts. After the third immunization, the antibody titer in the T/F and w4.26 groups were significantly higher than that in the BG505 group (*p* < 0.0001, *p* < 0.0001). After the fifth immunization, the antibody titers in the w4.26 group were significantly higher than those in the T/F group (*p* = 0.0368) and the BG505 group (*p* < 0.0001). Since the boost immunogens (Con_w22, Con_w30, Con_w53 and Con_w78 proteins) used in the T/F and w4.26 groups were identical, the higher titers of binding antibodies in the w4.26 group indicated that w4.26 priming could trigger stronger antibody responses. Overall, both CH505 Env groups induced higher titers of binding antibodies than those from the BG505 group. 

### 2.4. Elicitation of Broad Neutralization Activity

We next determined the neutralization activity of the sera from the vaccinated guinea pigs by assaying against the global panel of 12 tier 2 viruses (1 clade A, 2 clade B, 3 clade C, 1 clade AC, 1 clade G, 2 clade CRF01 and 2 clade CRF07) that are widely used to estimate the neutralization breadth and two easy-to-neutralize tier 1 viruses [53]. All immunogens induced similar neutralization titers against two tier 1 viruses (MW965 and SF162), with titers significantly higher across immunized groups for MW965 compared to SF162 (*p* = 0.002 for all groups, by Wilcoxon test) (Table A1). For tier 2 viruses, sera from guinea pigs in the T/F, w4.26 and BG505 groups neutralized 398F1, CH119 and CE1176 similarly well (Figure 4a). Other than these three viruses, the sera from the T/F or w4.26 groups neutralized many other viruses, while only a few sera from limited guinea pigs in the BG505 group neutralized three other viruses (X1632, 25710 and BJOX2000). 

When the neutralization potency against the panel of 12 tier 2 viruses representing most circulating clades was compared among three groups, the neutralization titers in the T/F group were significantly higher than those in the BG505 group (*p* = 0.0013). On average, all three groups neutralized 28–49% of the viruses. The T/F group neutralized most of the viruses (49%) while the BG505 group neutralized the least (29%). The neutralization breadth in the T/F group was significantly higher than that in the BG505 group (*p* = 0.0043; Figure 4a). The w4.26 group neutralized 43% of the viruses, but this was not statistically different from the T/F group or the BG505 group. 

Overall, when considering all 12 tier 2 viruses, neutralizing Ab responses were the weakest in the BG505 group compared to those in the T/F and w4.26 groups, both in magnitude and breadth, and the differences were statistically significant between the T/F and BG505 groups (*p* = 0.0013 and *p* = 0.0043, respectively), but not between the w4.26 and BG505 groups (Figure 4b,c). Since the CH505 and BG505 Envs were from clade C and A viruses, respectively, we excluded one subtype A and three subtype C viruses from the panel to determine the real cross-subtype neutralization activities. The magnitude of neutralization responses was lower against the 8-virus panel (excluding clade A and C viruses) than the full 12-virus panel among all three groups. The reduced potencies were significant for the T/F and BG505 groups (*p* = 0.002 and *p* = 0.02, respectively) but not significant for the w4.26 group (*p* = 0.18). The neutralization breadth was also significantly lower against the 8-virus panel than the full 12-virus panel among all three groups (Figure 4b,c). Although the potency and breadth of neutralization were all reduced when tested against only the more divergent clade-unmatched viruses, the sera from the T/F group still significantly better than those from the BG505 group both in potency and breadth (*p* = 0.001 and *p* = 0.004, respectively), but not significantly different from those from the w4.26 group (Figure 4b,c). 

We next selected seven tier 2 viruses (CH119, 246F3, X2278, 25710, BJOX2000, CE1176 and 398F1) that were frequently neutralized (more than two positive sera in each group) among three groups to determine the timing of development of neutralization activities. After third immunization, most sera from the T/F and BG505 groups could neutralize CH119 and 398F1, but only three sera from the BG505 had low tier neutralization activity against BJOX2000 (Figure 5). More sera had higher neutralization titers against 398F1 and CH119 after the fourth immunization than the third immunization. By this time, some sera from the T/F group gained the ability to neutralize X2278 and CE1176, while some sera from the BG505 group neutralized 246F3 but lost the low level of neutralization activities against BJOX2000 (Figure 5). After the fifth immunization, most sera from all three groups could neutralize many of these viruses, with the sera from the T/F group showing the highest potency and breadth among all three groups (Figure 5). These results indicate that neutralization activity was gradually increased after each sequential immunization. 

### 2.5. Different Neutralization Specificities among Immunogen Groups

To understand if the different immunogens and immunization strategies induce distinct neutralization responses, we compared the neutralization specificities of sera from the T/F, w4.26 and BG505 groups using three CH119 mutants which had the N88A mutation (gp120/gp41 interface), the N160K mutation (V2 apex) or the N279A mutation (CD4bs) since most sera from all three groups had high neutralization activity against CH119 (Figure 4; Table A1). When tested against mutant N88A, sera from both the T/F and w4.26 groups had greatly reduced neutralization titers (>70%), while no reduction in neutralization activities was observed for the sera from the BG505 group. These results indicated that neutralizing Abs with the gp120/gp41 interface specificity were elicited in the T/F and w4.26 groups but not in the BG505 group. When tested against mutant N160K, only a marginal reduction in neutralization activities were found in the sera from the T/F and w4.26 groups, while the neutralization titers were reduced about 10 folds (89.5%) in the sera from the BG505 group, indicating that that neutralizing Abs with the V2 specificity were elicited in the BG505 group but not in the T/F and w4.26 groups. When tested against mutant N279A, the neutralization titers were only weakly reduced (about half) in the sera from all three groups (Figure 6). These results suggest that the different immunogens and/or immunization strategies could induce antibody responses with distinct neutralization specificities; the nAb in the sera from the CH505 immunization group more likely target the gp120/gp41 interface epitope while the nAb in the sera from the BG505 immunization group more likely target the V2 epitope. 

## 3. Discussion

HIV-1 vaccines are notoriously difficult to develop. After 40 years of intensive research, a successful HIV-1 vaccine still remains to be developed. One of the main hurdles is the extraordinarily high level of genetic variations among HIV-1 genomes. Thus, the induction of broad reactive immune responses against diverse HIV-1 strains has been one of the major focuses in HIV-1 vaccine research. By mimicking the natural infection history in a person who has developed bnAbs [24], we used the stable UFO Env trimers based on the consensus sequences that best represented the quasispecies populations at critical time points to sequentially immunize guinea pigs and induced more potent and broader nAb responses than the gold standard BG505 Env trimer. 

Our previous studies of the co-evolution of bnAbs and viruses in an HIV-1-infected individual showed the diversification of viruses preceding the development of broad neutralization [24], indicating that immunization with the increasingly diversified HIV-1 Env sequences from such an individual is likely to recapitulate the maturation process of bnAbs in vivo [25]. However, the random selection of one Env sequence from a quasispecies population at a random time point and sequential immunization with such Env immunogens elicited limited nAb responses against heterologous tier 2 viruses in macaques [40,41]. This may be partly due to the arbitrary selection of Env sequences and time points potentially not recapitulating the process that induced bnAbs in the individuals. In addition, the early accumulated mutations were strongly selected by autologous nAbs and became dominant in the viral populations when the neutralization against heterologous tier 2 viruses was detected by week 78 [24]. These early mutants also played a critical role in driving the maturation of bnAbs in vivo [25]. However, these early Env mutants were not tested in previous studies [40,41]. Therefore, the use of stable UFO Env trimers generated from consensus sequences that can better represent the viral populations at the early time points (weeks 4, 22, 30, 53 and 78) for sequential immunization to better mimic the natural infection history may have allowed the elicitation of broader and more potent neutralization activities than the repeated immunization of the BG505 Env trimer.

Sequential immunization with subject CH505 consensus Env trimers from different time points preferentially induced gp120/gp41-interface-specific nAbs, while the repeated immunization of the BG505 Env trimer was more likely to elicit V2 specific nAbs. This demonstrates that different immunization strategies with disparate Env immunogens can elicit nAb responses with distinct specificities. However, more studies are required to investigate whether the consensus Env sequences, increasing diversification of the Env immunogens or mimicry of natural infection play the critical role in the induction of bnAbs with distinct specificities. 

The new UFO design shows the much improved stability and immunogenicity of the Env trimers [21]. The UFO Env trimer design can be easily applied to different HIV-1 Envs, but the expression levels can be highly variable among different Envs [22,54,55]. There is a positive correlation between the stability and immunogenicity of UFO Env trimers. A study shows that unstable proteins may be rapidly decomposed in the body and expose epitopes not for nAbs, while stable trimer protein can more likely display determinants targeted by bnAbs [52]. To further improve the expression levels, stability and immunogenicity of Env trimers, a second-generation UFO (UFO-BG) trimers were designed for diverse HIV-1 Envs by replacing their gp41 with BG505 gp41 based on the original UFO design [22]. Thus, the utilization of this more stable second-generation UFO strategy may induce even stronger immune responses in future studies.

Both the T/F and a natural variant w4.26 were from the first available week 4 sample, and they differ from each other by only one amino acid at position 501 [25]. Our early study showed that the affinity of w4.26 Env to autologous CH103 Ab was 10 times higher than T/F Env. Thus, we expected that w4.26, as the priming immunogen, could more effectively trigger the broad neutralization responses in guinea pigs. However, slightly lower neutralization potency and breadth were observed in the w4.26 group than the T/F group (*p* > 0.05). A cystine needs to be introduced at position 501 by replacing alanine to generate a disulfide bond in the UFO Env trimer design [21]. However, to preserve the unique sequence of w4.26, the valine at the same 501 position was not changed (Figure 2). Thus, the A501V mutation in w4.26 might not form the desired disulfide bond and affect the stability of the w4.26 Env trimers (Figure 3). This may explain why priming with the w4.26 trimer did not yield better neutralization responses than the T/F trimer, although it induced higher levels of binding Abs than the T/F and BG505 Env trimers. 

In summary, the approach to use consensus Env sequences from critical time points to generate stable Env trimers can induce more potent and broader nAb responses by mimicking natural infection history in the person who has developed bnAbs than the repeated immunization of the gold-standard BG505 UFO Env trimer. The unique features, like consensus Env sequences for each time point, stable UFO Env trimers and sequential immunization with the T/F Env (for triggering B cell precursors), as well as increasingly divergent Envs from different time points (for driving maturation of bnAb lineages), may be more likely to induce broad neutralization activity. This new approach demonstrates a great potential to improve nAb responses induced by HIV-1 vaccines. Continued optimization the Env trimer design such as by developing more stable UFO-BG trimers, Env trimers with a better exposed “up” position V2-apex [28] or nanoparticle vaccine approaches [56] may further improve the immunogenicity of the current approach.

## 4. Materials and Methods

### 4.1. Generation of Env Consensus Sequences for Different Time Points from Subject CH505

All available sequences from week 4 to week 78 from subject CH505 generated in our previous study [24] were aligned together based on the amino acid sequences using Codon Alignment (https://www.hiv.lanl.gov/content/sequence/CodonAlign/codonalign.html (accessed on 9 September 2022)). The alignments were optimized by manual adjustments, and a consensus sequence was generated for the virus from each time point using Seaview 5.0.5 [57]. We previously found that a natural variant (w4.26) that differed from the CH505 T/F Env sequence by one mutation (A501V) bound CH103 lineage bnAbs 10 times higher than the T/F Env and was also more sensitive to neutralization than the T/F virus [24]. To test whether the higher affinity of this w4.26 variant can bind and trigger the B cell precursors of CH103 lineage bnAbs more efficiently, the w4.26 Env protein was also used as a priming immunogen. The CH505 consensus Env sequences (T/F, Con_w22, Con_w30, Con_w53 and Con_w78) and the mutant w4.26 sequences were designed to express Env proteins as the stable UFO trimer form, as previously described [21]. All these Env sequences were codon-optimized and chemically synthesized (GenScript, Nanjing, China) for high-level expression in mammalian cells. The synthesized env genes were cloned into a eukaryotic expression vector phCMV3. All final plasmids were confirmed by sequencing.

### 4.2. Expression and Purification of UFO Env Trimer Proteins

HEK293-6E cells (NRC, Ottawa, Canada) were maintained in 125 mL Erlenmeyer flasks (NEST, Wuxi, China) in OPM-293 CD05 Medium (OPM Biosciences, Shanghai, China, Cat. 81075-001) supplemented with 0.1% Poloxamer 188 solution (Sigma-Aldrich, St. Louis, MO, USA, Cat. P5556-100ML) in a shaker incubator at 37 °C and 5% CO_2_ at 100 rpm. The cells were split to ensure that they were in the exponential growth phase on the day of transfection. CH505 UFO plasmid DNA (1 mg) in 20 mL of OPM-293 CD05 Medium was mixed with four milligrams of Polyethylenimine (PEI) Prime^TM^ (Sigma-Aldrich, Cat. 919012) in 20 mL of OPM-293 CD05 Medium and incubated for 30 min at 25 °C. The complex was then added into 800 mL HEK293-6E cells (1 × 10^6^ cells/mL). After 72 h, the cell supernatant was harvested and centrifuged at 1000× *g* for 2 h followed by 8000× *g* for 30 min. The expressed proteins were first purified by binding to the Lentil Lectin Sepharose™ 4B column (GE Healthcare, Boston, MA, USA, Cat. GE17-0444-01) and eluting with 500 mM methyl-α-D-mannopyranoside (Sigma-Aldrich, Cat. 462711). The purified UFO trimers were separated from other oligomers and monomers via size exclusion chromatography (SEC) on a Superdex 200 Increase 10/300 G L column (GE Healthcare, Cat. GE28-9909-44). The final purified UFO trimers were confirmed by blue native gel analysis using NativePAGE™ 3–12% Bis-Tris Protein Gels (Invitrogen, Carlsbad, CA, USA, Cat. BN1001BOX).

### 4.3. Western Blot Analysis

The culture supernatants or purified Envs from the transfected cells were analyzed using sodium dodecyl sulfate-native polyacrylamide gel electrophoresis (SDS-PAGE). The separated proteins were transferred onto a polyvinylidene difluoride (PVDF) membrane at 16 V for 16 min. The membrane was then blocked with 3% defatted milk for 30 min at 25 °C and probed with mAb 3B3 (1:10,000) overnight at 4 °C and then with Goat anti-mouse-HRP (1:10,000; Jackson, West Grove, PA, USA, Cat. 115-035-020) for 30 min at 25 °C. After final washes, the gel image was obtained using the Tanon 5200 series fully automated chemiluminescence/fluorescence image analysis system (Tanon, Shanghai, China). 

### 4.4. Immunization

A total of 24 female guinea pigs weighing between 200 and 250 g were randomly divided into 4 groups (six animals per group): T/F, w4.26, BG505 and PBS. In the T/F group, the guinea pigs were primed with the T/F UFO Env trimer, and then sequentially boosted with Con_w22, Con_w30, Con_w53 and Con_w78 UFO trimers as they appeared naturally in the infected individual CH505. To investigate whether the higher Env binding affinity and greater sensitivity to neutralization of one earlier variant, w4.26, could bind and trigger the B cell precursors more efficiently, the w4.26 Env trimer was used as the priming immunogen in the w4.26 group. The animals were then sequentially boosted with Con_w22, Con_w30, Con_w53 and Con_w78 UFO trimers as in the T/F group. In the BG505 group, the guinea pigs were repeatedly immunized five times with BG505 UFO trimers, while the guinea pigs in the PBS group were immunized with PBS five times. For each immunization, 50 μg of each immunogen (0.1 mL) was mixed with 0.1 mL of adjuvant AS03. Animals were immunized five times at a 3-week interval (Table 1). Blood samples were collected through heart puncture two weeks after each immunization, and the sera were harvested and stored at −80 °C until future use. All animals used in this study were handled according to the Guide for the Care and Use of Laboratory Animals (Changchun BCHT Biotechnology, Changchun, China). All experimental procedures were reviewed and approved by the Institutional Animal Care and Use Committee of BCHT (BCHT-AEEI-2019-015).

### 4.5. ELISA

Enzyme-linked immunosorbent assay (ELISA) was performed to determine HIV-1-specific IgG titers, as previously described [58] Briefly, ELISA plates (Jet BioFil, Guangzhou, China) were coated with CH505 T/F protein (100 ng/mL) per well in coating buffer (0.05 M carbonate-bicarbonate, pH 9.6) at 4 °C overnight. After three times of washing with washing buffer (1× PBS with 0.5‰ Tween-20), plates were incubated with 3% BSA at 37 °C for 2 h. After three times of washing, serum samples were serially diluted at a 1:10 ratio (from 1:100 to 1:100,000) and added to the plates (100 μL per well) in duplicates and incubated at 37 °C for 2 h. Then, the plates were washed three times and incubated with goat anti-guinea pig IgG-HRP (1:10,000) at 37 °C for 1 h. After three times of washing, 3,3′,5,5′-Tetramethylbenzidine (TMB) was added to the plates (100 μL per well), and the plates were incubated in the dark at 25 °C for 30 min. After incubation, 50 µL of H_2_SO_4_ (2 M) were added to the plates, and the absorbance was determined at 450 nm on an iMarkTM Microplate Reader (Bio-Rad, Hercules, CA, USA).

### 4.6. Neutralization Assays

The global panel of HIV-1 Env plasmids was obtained through the NIH AIDS Reagent Program (catalog no. 12670). Three mutations, N88A, N160K and N279A, were introduced into the CH119 env gene individually to generate three escape mutants [59]. Env-pseudoviruses were prepared by co-transfecting an Env-deficient backbone plasmid (pSG3ΔEnv) and an HIV-1 Env-expressing plasmid into HEK293T cells at a ratio 2:1 with Lipofectamine 2000 transfection reagent (Invitrogen, Cat. 11668-019). Pseudoviruses were harvested 72 h after transfection by centrifugation at 1000× *g* for 20 min and stored at −80 °C until use. 

Neutralization assays were determined in 96-well culture plates using a luciferase reporter gene system on TZM-bl cells, as previously described [60]. The guinea pig sera were heat-inactivated (56 °C for 1 h), serially diluted at a 1:3 ratio (starting from 1:30) and incubated with the pseudovirus at 37 °C for 1 h. TZM-bl cells were then added, and the culture was maintained at 37 °C for 48 h. The cells were washed with PBS once and lysed in Reporter Lysis Buffer (Promega, Madison, WI, USA, Cat. E4030). Luciferase activity was measured on a Victor X Multilabel Readers (PerkinElmer, Waltham, MA, USA). Only the ID_50_ titers that were two times higher than the lowest dilution (1:30) were analyzed to avoid influence from low- and unstable low-neutralization titers (between 1:30 and 1:59).

### 4.7. Statistical Analysis

Statistical analysis was performed using one-way analysis of variance (ANOVA) to compare the differences in t values between the experimental and control groups and Student’s t-test in GraphPad Prism 9.0 (GraphPad, San Diego, CA, USA). *p* < 0.05 was considered significant. All experiments were repeated in triplicate. Values were expressed as the mean ± standard deviation (SD). The comparison was considered statistical significant when the *p* value is smaller than 0.05, and then correspondingly marked by asterisks in the figures, where * *p* < 0.05; ** *p* < 0.01, *** *p* < 0.001; **** *p* < 0.0001; and ns means non-significant.

For each animal, breadth was measured as the number of viruses (out of a total of 12) that yielded above threshold (≥60) ID_50_. Breadth between vaccine groups was compared using a 2-way Wilcoxon test. Magnitude of responses between groups was tested using a random effect generalized linear model (GLM) with vaccine groups as fixed effect and virus as random effect. Tests and models were run including all tier 2 viruses first and then excluding clade A and C viruses to exclude within-clade bias. Tests were considered significant when they yielded a false discovery rate (FDR) q-value of 0.5 or lower after multiple testing correction. All statistical tests were performed in R (https://www.R-project.org/ (accessed on 23 March 2023) using the base and lmer4 packages [60]. 

## Figures and Tables

**Figure 1 ijms-24-12642-f001:**
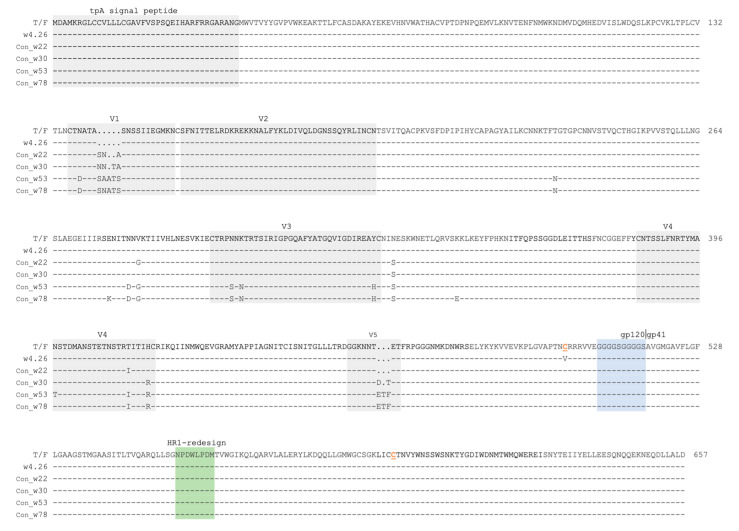
Comparison of amino acid sequences between the T/F, an early variant and consensus sequences for different time points. All sequences designed to express UFO Env trimers were compared to the CH505 T/F sequence. The identical amino acids are shown as dashes, and deletions are indicated by dots. The variable regions (V1 through V5) are indicated by gray shaded areas. The amino acids associated with disulfide bond (A501C/T605C) in the UFO Env trimer are underlined. The G4S linker is indicated by a blue shaded area, and the HR1-redesigned region is indicated by a green shaded area. The gp120 region and gp41 regions were separated by a vertical line.

**Figure 2 ijms-24-12642-f002:**
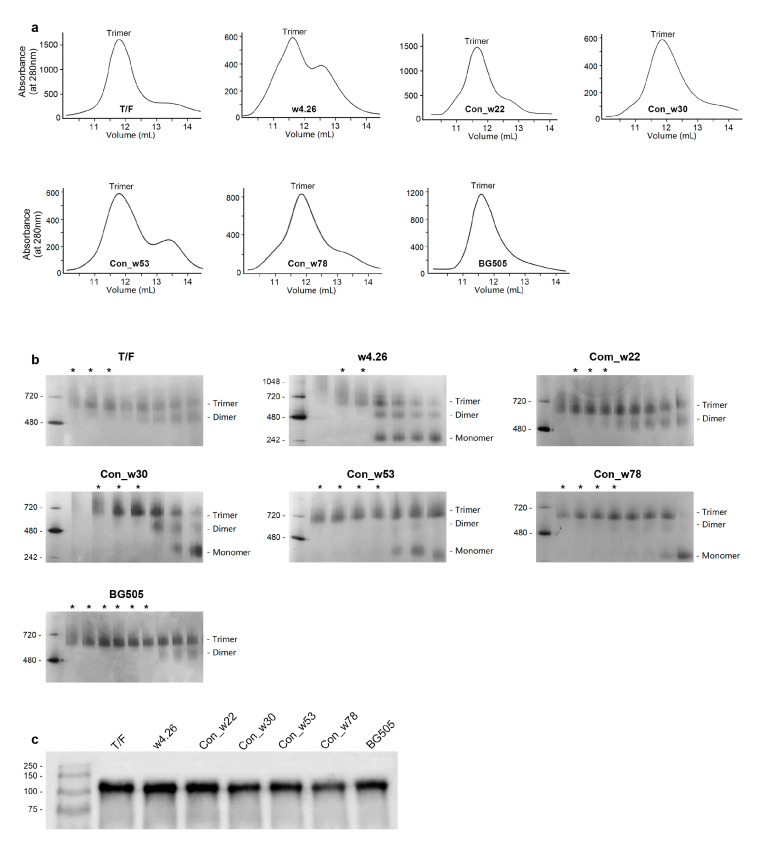
Characterization of UFO Env trimer proteins. (**a**) Size-exclusion chromatography (SEC) profiles of lectin-purified UFO proteins. The lectin-purified Env proteins were separated using gel filtration chromatography on a Superdex 200 10/300 column. Peak fractions were pooled and further analyzed using SDS-PAGE. (**b**) Blue native gel analysis of Env proteins collected from SEC. The fractions used for immunization are indicated by asterisks on the top of the gel. (**c**) Analysis of SEC-purified trimeric proteins by Western blot.

**Figure 3 ijms-24-12642-f003:**
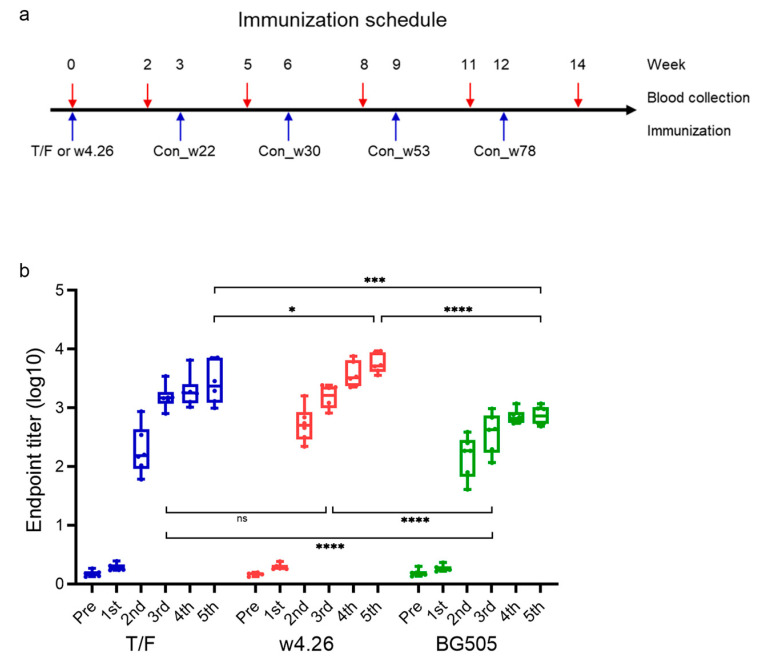
Comparison of HIV-1-specific antibody titers among different immunization groups. (**a**) Vaccination regimen for guinea pigs. Guinea pigs (*n* = 6) were randomly assigned into four groups and each guinea pig was immunized 5 times at a 3-weeks interval. The two immunization groups were primed with the T/F Env or w4.26 Env and then boosted with four different consensus Env trimers: Con_w22, Con_w30, Con_w53 and Con_w78. (**b**) Analysis of HIV-1-specific binding antibody titers in the guinea pig sera from different immunization groups. Each antibody titer for individual animal is indicated by a dot. Box and whiskers with median, from min to max. ns: non-significant, * *p* < 0.05; *** *p* < 0.001; **** *p* < 0.0001.

**Figure 4 ijms-24-12642-f004:**
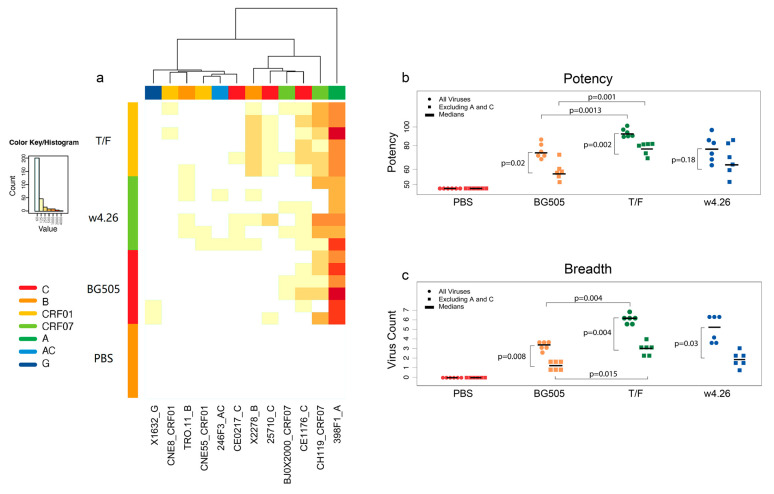
Comparison of neutralization breadth and potency among different immunogens. (**a**) Analysis of neutralization activities of the guinea pig sera. Sera were collected 2 weeks after the fifth immunization, and the neutralization was carried out against the virus panel clustered by clade. The 50% inhibitory doses (ID_50_) against the global panel of 12 tier 2 viruses were analyzed via heatmap. The neutralization potency is shown in different shades of colors, as indicated in the histogram. The ID_50_ titers equal or greater than 1:60 were considered positive and analyzed. (**b**) Comparison of neutralization potency. The geometric mean of ID_50_ was determined for the full 12-virus panel (circle) and the 8-virus panel (excluding clade A and C viruses; square) for each animal. Medians of the geometric means of ID_50_ are shown as black horizontal bars. Statistical comparisons among vaccine groups were conducted using a random effect generalized linear model. (**c**) Comparison of neutralization breadth. The numbers of neutralized viruses determined for each animal for the full 12-virus panel (circle) and the 8-virus panel (excluding clade A and C viruses; square). Medians are shown as black horizontal bars. Statistical comparisons among vaccine groups were done using 2-way Wilcoxon tests.

**Figure 5 ijms-24-12642-f005:**
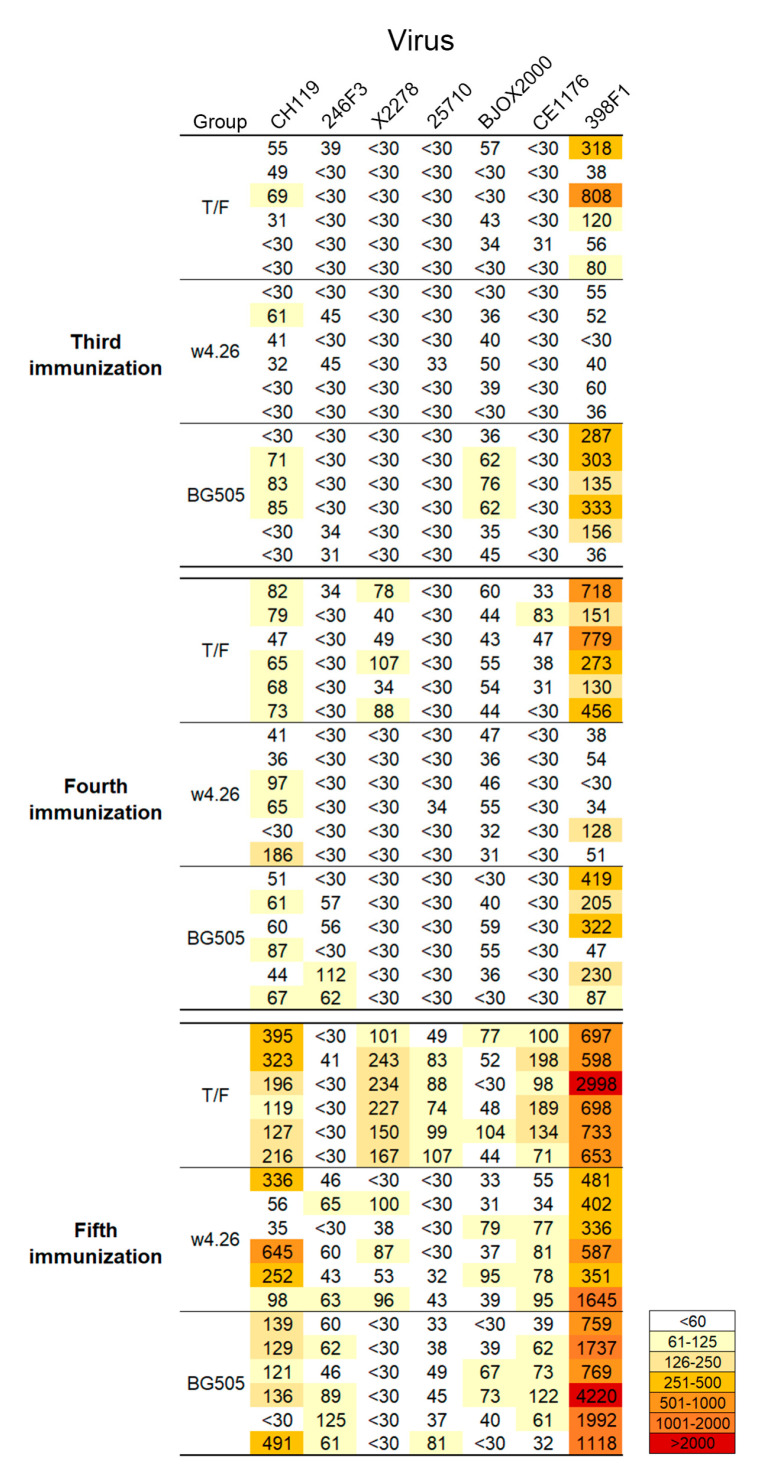
Development of neutralization breadth after each immunization. Sera from three groups (T/F, w4.26 and BG505) from the third immunization to the fifth immunization were assayed against the seven global panel viruses that were frequently neutralized (more than two positive sera in each group) among three groups. The neutralization titers (ID_50_) are shown in different shades of colors as indicated in the histogram.

**Figure 6 ijms-24-12642-f006:**
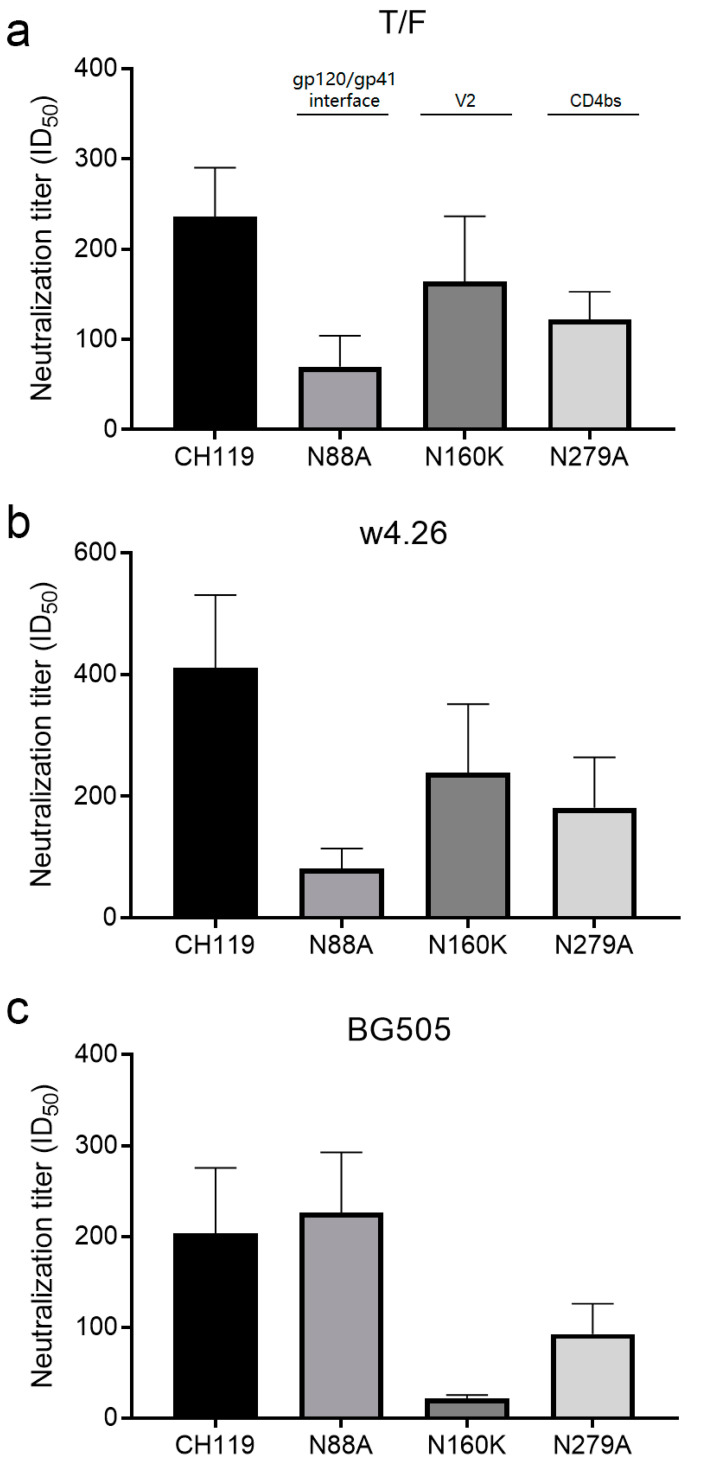
Neutralization specificities of sera from guinea pigs. Neutralization activity of sera after the fifth immunization from three groups T/F (**a**), w4.26 (**b**) and BG505 (**c**) were determined against the wild-type CH119 and its mutants (N88A, N160K, N279A). Data are presented as mean ± SEM (*n* = 5). Statistical comparisons of different vaccine groups were performed using one-way analysis of variance (ANOVA).

**Table 1 ijms-24-12642-t001:** Immunization groups and schedule.

Group	Week 0	Week 3	Week 6	Week 9	Week 12
T/F	T/F	Con_w22	Con_w30	Con_w53	Con_w78
w4.26	w4.26	Con_w22	Con_w30	Con_w53	Con_w78
BG505	BG505	BG505	BG505	BG505	BG505
PBS	PBS	PBS	PBS	PBS	PBS

## Data Availability

The original data are available upon reasonable request to the corresponding author.

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
