# Peer review of "Enhancement of Neutralization Responses through Sequential Immunization of Stable Env Trimers Based on Consensus Sequences from Select Time Points by Mimicking Natural Infection"

_ijms, 2023, doi:10.3390/ijms241612642_

Round 1

Reviewer 1 Report

This paper by Wan et al. follows on with studies  of the HIV-1 BG505 Env trimer and employs sequential immunization of guinea pigs with priming by BG505, the T/F Env from that patient, or an Env w4.26 that has only one mutation (A501V) compared to the T/F. The priming is then follows by sequential immunizations using BG505 Env in the BG505 primed group, or using consensus Envs from weeks 22, 30, 53, 78 in the other 2 groups. The binding antibodies are improved in the groups primed with the T/F and w4.26 followed by immunizations with the patient Envs, and the Nabs are somewhat improved in those 2 groups.  However, it's not clear if the different primes are most responsible for this or if the sequential boosting with multiple different patient Envs is most responsible. A group that was primed with BG505, but then sequentially immunized with Envs from weeks 22/30/53/78 may have shed light on this question.

That aside, the data are an important next step in sequential immunization strategies attempting to mimic natural infection. There is a modest improvement in antibody responses in the non-BG505 2 groups and this may be important and should be conveyed to the literature. In the pdf, the important figure 4 color key and histogram is fuzzy and it would be best to simply use the tradition white-yellow-orange-red heat map traditionally used in the field with a legend that indicates titer ranges for each color of the heat map.

The use of the word "Critical" in the title is somewhat misleading and might be replaced with "selected diverging time points" or something of that nature. 

There are multiple minor errors in English grammar throughout the paper in numerous locations to include tense, use of plural, pronoun usage etc but these errors will be very simple to correct by publication editors.

Reviewer 2 Report

The authors generated a consensus Env vaccine based on sequences obtained from a subject who developed broad neutralization antibody. The authors sequentially immunized guinea pigs with different Env vaccines and antibody responses were analysed by ELISA and neut assay. The result looks promissing. The manuscript is well written and the experimental approaches as well as the conclusions are clearly described. Comments for the authors below:

Major points:

1.      Line 363: Please indicate the source of reagents (Company name, Cat. # etc) so that the readers can follow your methods.

2.      Figure 6: Please explain error bars and include stats.

3.      Please explain why T-cell immunity was not measured.

Minor points:

1.      Please make sure the Figure numbers. “Fig. A1” should be “Fig. S1” and so on (A2, A3).

2.      Line 206: “Supplementary table 1” should be “Table A1 or S1”.

3.      Line 437: Please replace xxx by number.
